# Biological and health-related effects of weak static magnetic fields (≤ 1 mT) in humans and vertebrates: A systematic review

**Sarah Driessen\*, Lambert Bodewein, Dagmar Dechent[ID], David Graefrath, Kristina Schmiedchen, Dominik Stunder, Thomas Kraus, Anne-Kathrin Petri[ID]**

Research Center for Bioelectromagnetic Interaction, RWTH Aachen University, Aachen, Germany

\* driessen@femu.rwth-aachen.de

**Data Availability Statement:** All relevant data are within the paper and its Supporting Information files.

## Abstract

### Background

There is a rapid development in technologies that generate weak static magnetic fields (SMF) including high-voltage direct current (HVDC) lines, systems operating with batteries, such as electric cars, and devices using permanent magnets. However, few reviews on the effects of such fields on biological systems have been prepared and none of these evaluations have had a particular focus on weak SMF (≤ 1 mT). The aim of this review was to systematically analyze and evaluate possible effects of weak SMF (≤ 1 mT) on biological functioning and to provide an update on the current state of research.

### Methods

This review was prepared in accordance with the PRISMA (Preferred Reporting Items for Systematic Reviews and Meta-Analyses) statement. Methodological limitations in individual studies were assessed using the Office of Health Assessment and Translation (OHAT) Risk-of-Bias Rating Tool.

### Results

Eleven studies fulfilled the eligibility criteria and were included in this review. All included studies were experimental animal studies as no human studies were among the eligible articles. Eight of the eleven studies reported responses of rat, rabbits and quails to weak SMF exposure that were expressed as altered melatonin biosynthesis, reduced locomotor activity, altered vasomotion and blood pressure, transient changes in blood pressure-related biochemical parameters, or in the level of neurotransmitters and increases in enzyme activities. It remained largely unclear from the interpretation of the results whether the reported effects in the evaluated studies were beneficial or detrimental for health.

### Conclusion

The available evidence from the literature reviewed is not sufficient to draw a conclusion for biological and health-related effects of exposure to weak SMF. There was a lack of

**Funding:** This project was funded by the Federal Ministry of Education and Research, Germany, Forschungscampus Elektrische Netze der Zukunft (FKZ03SF0495). The funders had no role in study design, data collection and analysis, decision to publish, or preparation of the manuscript.

**Competing interests:** The authors have declared that no competing interests existed during the conduct of the study. However, during the revision of the document, DS has become an employee of Amprion GmbH, a German grid operator.

homogeneity regarding the exposed biological systems and the examined endpoints as well as a lack of scientific rigor in most reviewed studies which lowered credibility in the reported results. We therefore encourage further and more systematic research in this area. Any new studies should particularly address effects of exposure to SMF on biological functioning in humans to evaluate whether SMF pose a risk to human health.

## 1 Introduction

More recently, the installation of new high-voltage power lines [1, 2] and the introduction of novel technologies that produce electromagnetic fields (EMF), such as 5G networks [3], smart meters [4], and electric vehicles [5] have led to controversial discussions among the public, politicians, non-governmental organizations, and the industry about the benefits of these technologies and the possible risks of exposure to non-ionizing radiation. Besides environmental impacts and legal requirements, the discussions have often centered on potential hazards to health from EMF. Public opposition often delays the roll-out of new technologies threatening the economy on various scales due to, e.g., delays in technical progress, potential international competitive disadvantages, and serious financial losses. In Germany, both the launch of 5G networks and the plans for the build-out of the power grid, including four cross-country high-voltage direct current (HVDC) power lines, have attracted growing public interest. New power lines are planned to transfer power generated by remote renewable sources (particularly wind power) to areas where the demand for energy is high. HVDC power lines can transport electricity over long distances with lower line losses compared to conventional alternating current power lines [6]. Although the intensity of static magnetic fields (SMF) emitted from HVDC lines is comparatively weak, there is public concern about possible health-related and environmental impacts of SMF produced in the vicinity of HVDC power lines.

SMF are constant fields, i.e., they do not change in intensity or direction over time, and thus have a frequency of 0 Hz. The intensity of magnetic fields is described by the magnetic flux density, usually expressed in tesla (T) for static fields. SMF can be classified as weak ($< 1$ mT), moderate (1 mT to 1 T), high (1 T to 20 T) and ultra-high ($> 20$ T) [7–9]. All living organisms, including humans, are continuously exposed to a natural SMF–the geomagnetic field (GMF)–ranging from approximately 25 µT at the equator to 65 µT at the poles [10]. In addition to HVDC lines, other man-made SMF generating sources are technical devices using direct current (DC) or permanent magnets (e.g., electric cars, loudspeakers, microphones) or medical devices such as magnet resonance imaging (MRI). HVDC lines produce weak SMF between 22 µT and 38 µT in close proximity [6, 11, 12]. For hair dryers, stereo headsets, home sewing machines, and electric clocks, it has been documented that their geometric mean SMF are between 50 µT and 93 µT, depending on measurement distance and location [13]. Weak SMF were measured inside hybrid technology cars (up to 0.95 mT) [14] and inside the driver's cabin in DC trains (about 1 mT) [15]. Moderate SMF may be produced by magnetic levitation train systems (up to 10 mT) [16] and in certain locations of aluminum production plants (60 mT) [17]. Under common exposure scenarios, MRI workers are exposed to SMF in the order of several hundreds of mT [18, 19] or even 2,890 mT (7 T scanner) in a research environment [20].

It is known that SMF–in contrast to static electric fields–can penetrate living systems and directly interact with moving electric charges (e.g., ions) through several mechanisms [17]. The available data on the effects of SMF on biological functioning in humans and animals

have been analyzed and evaluated in various reviews and governmental environmental impact assessments. The most commonly reported effects of exposure were transient symptoms such as vertigo, nausea, magnetic phosphenes, and a metallic taste in the mouth [17, 21–24]. These effects were observed for higher magnetic flux densities in the tesla range. They may occur when a person moves within a strong SMF but also due to blood flow when a resting body is subjected to a SMF. The induced electric potentials and currents due to the blood flow result in a magnetohydrodynamic force [23, 25]. Additionally, previous research suggests that weak SMF may act on electron spin interactions [8, 17, 22, 23] and that humans can sense SMF in the range of the GMF [26, 27].

Exposure to SMF below the international limit values recommended by the International Commission on Non-Ionizing Radiation Protection (ICNIRP) [22], i.e., 400 mT for the public, is considered safe and it is not expected to pose a risk to health. However, the proposed exposure limits to electric, magnetic, and electromagnetic fields are based on short-term (acute) health effects. This has led to strong public and scientific debate about the sparsely investigated effects of long-term exposure to low intensity EMF for the population. EMF sources that have attracted most attention were mobile phones and alternating current power lines (50/60 Hz) because it was discussed that long-term exposure to low intensity radiofrequency or extremely low frequency magnetic fields may increase the risk of cancer [28, 29].

The aim of this systematic review was therefore to evaluate whether there is evidence that weak SMF ($\leq$ 1 mT), as they are produced, e.g., near HVDC lines and other man-made SMF sources of our daily life, can affect biological functioning in humans and vertebrates and cause adverse health effects. Formerly published reviews and reports on possible health risks from exposure to SMF were not conducted systematically and little information has been provided regarding the effects of exposure to weak SMF ($\leq$ 1 mT). However, SMF sources that produce magnetic flux densities below 1 mT are the most relevant sources for public exposure. We collected and analyzed experimental *in vivo* studies on short-term and long-term biological and health-related effects of exposure to weak SMF. Our review is intended to critically appraise the internal validity of the published evidence, identify open research questions, and support risk communication activities on the potential hazards of EMF. Although the likelihood of adverse health effects from exposure to weak SMF below the limit values was judged to be low, with the growing exposure to SMF produced by technical applications there is public and scientific demand for periodic evaluations of the current state of research in order to reassess and confirm the safety of weak SMF as they occur in daily life.

This systematic review constitutes the third part of a series of comprehensive literature analyses that assess the potential for adverse effects of static magnetic and static electric fields. Our previously published systematic reviews evaluated biological and health-related effects of exposure to static electric fields in humans and vertebrates [30] and in invertebrates and plants [31].

## 2 Methods

The "Preferred Reporting Items for Systematic Reviews and Meta-Analyses" (PRISMA) were used to guide the methodological conduct and the reporting of this systematic review [32]. To assess the internal validity of individual studies, we used the risk-of-bias rating tool recommended by the National Toxicology Program (NTP) [33].

### 2.1 Eligibility criteria

Eligibility criteria were determined using the Participants/Population (P), Exposure (E), Control (C), Outcome measures (O) (PECO) strategy [33]. Eligible for this review were

experimental *in vivo* studies on humans or vertebrates (P) with exposures to man-made SMF $\leq$ 1 mT (E). To be further eligible for inclusion, exposure groups had to be compared to a non-exposed control group or a sham exposure condition (C). For practicability, we only considered studies in which the SMF exposure level was higher in the experimental group/condition than in the control group/condition (GMF and background field) such that the GMF was sufficiently controlled as a possible confounder. Therefore, a magnetic flux density also had to be provided for the control group/sham condition. Furthermore, there was no restriction with regard to the examined endpoints, i.e., any outcome measures of biological or health-related effects were considered (O).

Articles had to be written in English or German and there was no restriction with regard to the year of publication.

Excluded were review articles, editorials, commentaries, unpublished, or non peer-reviewed articles as well as studies on simulations and dosimetric or theoretical aspects. Also excluded were studies with co-exposures (e.g., as in MRI, which is a combination of exposures to SMF, radiofrequency EMF and gradient magnetic fields or as in aluminum reduction plants, which are subject to multiple exposures such as heat and chemicals), because in these studies it may not be possible to separate a potential effect of exposure to SMF from the effects of the other exposure types.

Studies examining the influence of a geomagnetic storm or geomagnetic disturbances on health-related effects were excluded because they mainly investigate fluctuations of the GMF in the nanotesla range or experimentally simulate these fluctuations. The results of these studies preclude dosimetric considerations between magnetic flux densities and potential effects of exposure because the effect may be caused by the fluctuation itself rather than by a specific magnetic flux density. For this reason, these studies are not relevant for our systematic review.

Similarly, studies investigating an attenuated or hypomagnetic field such as, e.g., in a space environment, were not within the scope of our review. Because living organisms, including humans, are continuously exposed to the natural GMF, our research question was focused on man-made SMF that are superimposed on the natural GMF and not on the attenuation of the GMF. It has to be noted, however, that under specific circumstances the natural GMF may be subject to attenuation by man-made SMF.

As many species are able to perceive and orient to the GMF, magnetoreception and magnetic sensitivity were examined in a large number of studies. A great many of these studies investigated the effect of a variation in the magnetic flux density on magnetoreception or varied the inclination angle, the polarity, or other environmental cues, such as light parameters. Because of their particular focus, studies on magnetoreception merit a separate evaluation and were therefore excluded from our review.

## 2.2 Information sources and literature search strategy

Relevant articles published through October 2019 were identified through electronic searches in PubMed (U.S. National Library of Medicine, National Institutes of Health) and in our highly specialized literature database EMF-Portal (www.emf-portal.org). The EMF-Portal is the most comprehensive scientific literature database on biological and health-related effects of EMF with an inventory of currently 30,260 publications (January 2020). It was approved by the World Health Organization (WHO) as a reference database (https://www.who.int/peh-emf/research/database/en/). An independent evaluation in 2017 proved a completeness of the relevant literature in the EMF-Portal of more than 97% [34]. The identification of relevant studies to be included in the EMF-Portal is based on systematic search strategies in major databases, including PubMed, Cochrane Library, and IEEE Xplore Digital Library. The searches

are conducted periodically, i.e., these databases are screened daily or at least weekly. To supplement the electronic database searches, additional records are identified by screening scientific journals not listed in these databases and reference lists of journal articles and reviews. Prior to inclusion in the EMF-Portal, all records are labeled as to study design (e.g., experimental, epidemiological), exposure specifications (e.g., SMF, intermediate frequency or Wi-Fi), main endpoint(s) (e.g., cell proliferation, brain activity, genotoxicity) and type of publication (e.g., original research article, review, editorial, commentary). This *a priori* categorization enables us to perform highly specialized searches and ensure best search results.

The search terms were related to exposures and included the following key words: static magnetic field(s), DC magnetic field(s), constant magnetic field(s), stationary magnetic field (s), steady magnetic field(s), magnetostatic field(s), high-voltage direct current, HVDC. The search strings and links to the electronic databases are provided in the S1 Link (search string).

### 2.3 Study selection

The studies were screened for eligibility in two stages based on the eligibility criteria. In the first stage of assessment, the titles and abstracts of the identified articles were screened independently by two authors (AP, SD, LB, or DD). Articles which failed to meet the inclusion criteria were excluded. In the second stage of assessment, the full texts of potentially eligible articles were retrieved and independently reviewed by two authors (AP, SD, LB, or DD). The authors jointly made a final decision about the inclusion of relevant articles. Potential disagreements were discussed and resolved by consensus between the review authors.

### 2.4 Data extraction

Two authors (AP, SD, LB, or DD) independently extracted the data from the articles that met our eligibility criteria. The extraction protocol was defined and agreed upon before the start of the project. The extracted data included bibliographic data (e.g., author names, year of publication, journal), the exposed species, number and sex (if provided) of the examined individuals, the magnetic flux densities applied in the exposure group/condition and the control group/ condition, the exposure duration(s), the examined endpoints, and outcomes. Additional remarks were made about specific parameters (e.g., study background, additional experiments that were not relevant for this review) and about inconsistencies and particular limitations of individual studies. Disagreements and technical uncertainties were discussed and resolved between the review authors (AP, SD, DS, LB, DD).

### 2.5 Study appraisal

The internal validity (i.e., the degree to which the design, conduct, and analysis of a study avoid bias) and the overall risk-of-bias of the included studies was assessed by using the approach recommended by the NTP's Office of Health Assessment and Translation (OHAT) [33, 35]. The same rating tool was used in our previously published systematic reviews [30, 31, 36]. The OHAT risk-of-bias rating tool consists of a set of questions and provides detailed instructions on how to evaluate methodological rigor in human and animal studies with a focus on environmental health and toxicology. As recommended by OHAT, nine methodological criteria were applied to rate the included experimental animal studies for biases in selection, performance, detection, attrition/exclusion, or selective reporting. At least two authors (AP, SD, LB, or DD) independently assessed the risk-of-bias criteria for all included studies according to the following ratings: "+ +" definitely low risk of bias, "+" probably low risk of bias, "-" probably high risk of bias, or "- -" definitely high risk of bias. Potential disagreements between the authors were discussed and resolved by consensus.

To reach conclusions about the overall risk-of-bias of individual studies, we used the OHAT approach for categorizing studies into tiers (see S1 Table for details). This approach outlines a 3-tier system to rate study quality (1st tier: high confidence in the reported results, 2nd tier: moderate confidence in the reported results, or 3rd tier: low confidence in the reported results). Depending on the scope of the systematic review and the research question, OHAT suggests the definition of "key" risk-of-bias criteria. The "key" risk-of-bias criteria which were given the highest weight in determining the overall risk-of-bias within the scope of our evaluation were (1) "Were experimental conditions identical across study groups?", (2) "Can we be confident in the exposure characterization?", and (3) "Can we be confident in the outcome assessment?". The remaining risk-of-bias criteria were given less weight. Placement of a study into one of three study quality categories (1st tier, 2nd tier, or 3rd tier) was contingent on the rating of these three key risk-of-bias criteria and the proportions in the rating of the remaining criteria.

## 3 Results

### 3.1 Study selection

The systematic search returned a total of 5,712 articles (Fig 1). After removal of duplicates, 4,564 articles were screened in title and abstract, whereof 4,230 studies were excluded because they did not match the eligibility criteria (e.g., other type of experimental study such as *in vitro* studies or dosimetric studies, not EMF/health-related, review, editorial, comment, language not English or German, magnetic flux density > 1 mT). Note, that most of these studies were excluded because they met more than one exclusion criteria. For reasons of clarity, in the flow diagram, we only documented the most striking reason for their exclusion (Fig 1). The full text was obtained for the remaining 334 articles in order to check for the eligibility for inclusion in our analysis. Of these, 323 articles were excluded for several reasons: magnetic flux density > 1 mT (n = 162), studies on (mechanisms of) magnetoreception (n = 58), other type of experimental study (n = 22, e.g., *in vitro* studies, studies on invertebrates or plants, dosimetric approach), magnetic flux density not provided (n = 17), magnetic flux density of the control group/condition not provided (n = 15), exposure conditions unclear (n = 8), or investigation of field deprivation/hypomagnetic field (n = 8). Other articles were excluded because they investigated co-exposures (n = 7), MRI (n = 6), a geomagnetic storm condition (n = 5), or because they had no SMF exposure condition (n = 5). Reviews, editorials, comments (n = 6), an article not written in English or German (n = 1), a non EMF health-related article (n = 1), and a non peer-reviewed study (n = 1) were also excluded. One further article had to be excluded because it lacked the description of the results for the exposure groups. A list of all excluded articles including the bibliographic data and the reasons for their exclusion is provided in the Supplementary data (S2 Table). Finally, eleven studies fulfilled the eligibility criteria and were included in this systematic review.

### 3.2 Study appraisal

The internal validity of the included studies was assessed by using the risk-of-bias rating tool recommended by OHAT [33, 35]. Three of the reviewed studies [37–39] were placed in the "1st tier", the remaining eight studies were assigned to the "2nd tier" (see Fig 2).

Five studies adequately addressed all three key risk-of-bias criteria. In the remaining six studies, methodological flaws were mainly identified regarding two key criteria: Five studies lacked information on procedures to ensure *Identical experimental conditions across study groups* [40–44] and *Confidence in the outcome assessment* was lowered in three studies [40–42] due to, e.g., the use of potentially insensitive instruments. *Confidence in the exposure*

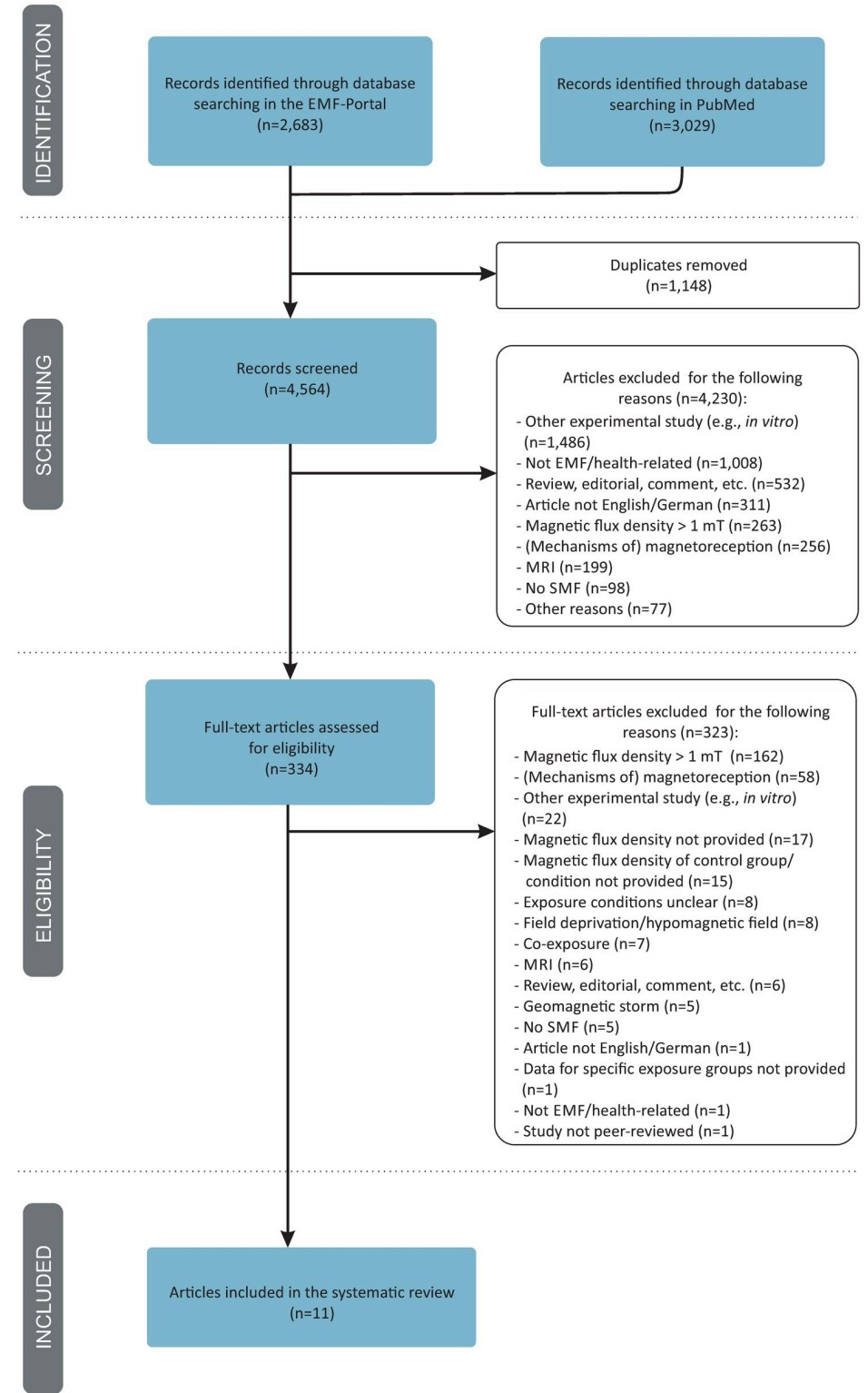

**Fig 1. Flow diagram of literature search, eligibility and inclusion process.** Adapted from Moher 2009 [32].

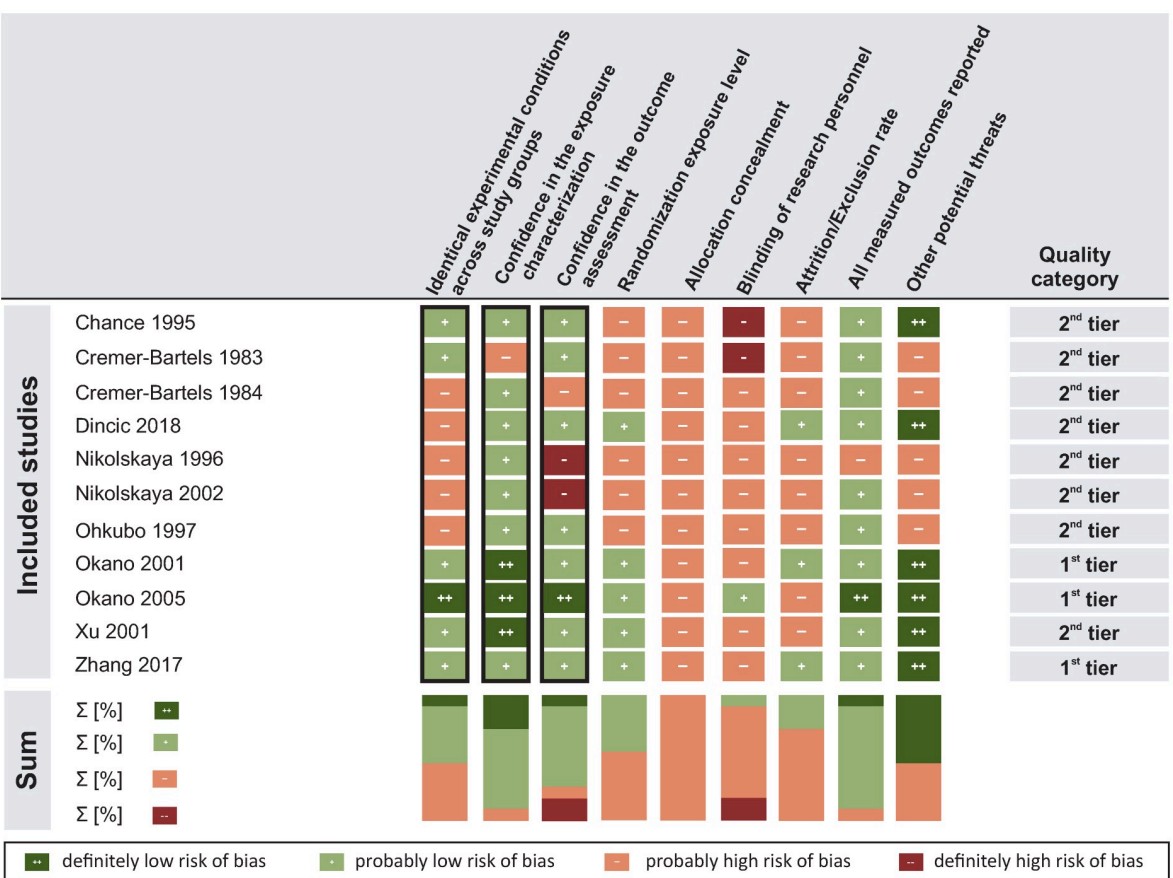

**Fig 2. Risk of bias in individual studies and across studies.** Black frames highlight key risk-of-bias criteria.

*characterization* was not assured in one study [45], while all other studies used valid and reliable methods to measure or simulate the intensities of the applied SMF exposures.

A number of potential threats to internal validity were also identified for the remaining risk-of-bias criteria. Methodological flaws that were common across studies were related to *Allocation Concealment*, *Blinding of the Research Personnel*, and *Attrition/Exclusion Rate*. All studies lacked information on whether the allocation of the animals to study groups was blinded. Furthermore, only one study was explicitly performed under blinded conditions during the experimental procedures while in the remaining studies, blinding of the research personnel was either not adequately addressed or no information on the blinding procedure was provided. Also, eight out of eleven studies did not provide sufficient information regarding the attrition/exclusion rate of data and/or animals which challenges the completeness of data and can thus be considered a risk of bias in these studies.

### 3.3 Results of individual studies

All of the evaluated studies were experimental animal studies as no human studies matched our eligibility criteria (see Table 1 for details). Various endpoints, including effects of exposure on melatonin biosynthesis [40, 45], behavior [41, 42], cardiovascular parameters [37, 38, 44, 46], and brain and nervous system [39, 43, 47], were studied in rodents or birds. The size of the experimental groups varied between 3 and 50 animals and the applied magnetic flux densities ranged from 52 μT to 1 mT.

**Table 1. Characteristics of individual studies (n = 11).**

| First author (Year) | Exposed animals (sex) Relevant groups (number of animals) | Exposure | Endpoints | Outcome | Remarks |
|---|---|---|---|---|---|
| **Melatonin biosynthesis** | | | | | |
| Cremer-Bartels 1983 [45] (2nd tier) | quails (male) exposure group: n = 3 (acc. to text) or n = 8 (acc. to figure) control group: n = 3 (acc. to text) or n = 8 (acc. to figure) | exposure group: SMF of approx. 52 μT* for 1 h control group: SMF of approx. 48 μT* (kept at a distance of 3.5 m away from coils) for 1 h axis of the coils was aligned in a horizontal North/South direction quails were exposed in groups of 3 animals | enzyme activity of hydroxyindole-O-methyltransferase (HIOMT) in pineal gland and retina | no statistically significant differences | also experiments with humans and additional experiments with quails (not relevant for this review) *magnetic flux density calculated from the provided data for single GMF components **study background** general interest whether SMF may influence biological systems |
| Cremer-Bartels 1984 [40] (2nd tier) | quails (sex not provided) n = 6 (unclear whether this is the total number of individuals in the exposure and the control group or the number of individuals in each group) | exposure groups: SMF of 72 μT* for 20 or 60 min control group: SMF of 48 μT* for the same durations axis of the coil was aligned along the GMF lines in the experimental area quails were exposed in groups of 3 animals | enzyme activity of hydroxyindole-O-methyltransferase (HIOMT) in the pineal gland | decrease of HIOMT enzyme activity after 20 (p≈0.025) and 60 min (p≈0.01) | also experiments with reduced SMF intensities and with chicken tissues (in vitro) (not relevant for this review); unclear whether measurements of enzyme activity of serotonin-N-acetyltransferase were in vivo or in vitro experiments (therefore not included in this review) *magnetic flux density calculated from the provided data for single GMF components **study background** circadian rhythm |
| **Behavior** | | | | | |
| Nikolskaya 1996 [42] (2nd tier) | Wistar rats (male) exposure group: n = 20 control group: n = 50 | exposure group: SMF of 55–240 μT for up to 13 min on 13 consecutive days control group: ambient SMF of 37 ± 2 μT for up to 13 min on 13 consecutive days rats were exposed during daily learning session in the maze but could leave the maze earlier and enter the open field SMF generated by loudspeaker magnets installed beneath the maze rats were exposed and examined individually in the maze | food-operant behavior in a complicated problem environment (maze) and behavior in an open field test | locomotor and emotional depression (p < 0.05) in the maze such that the exposed rats were unable to form food-operant behavior (showed impaired learning abilities); no statistically significant differences in the open field test | rats were treated differently: some rats in the exposure group received a "push" by the experimenter when motor activity inhibition occurred; also experiments with other magnetic flux densities (not relevant for this review) **study background** cognitive abilities |
| Nikolskaya 2002 [41] (2nd tier) | Wistar rats (male) exposure group: n = 50 control group: n = 50 | exposure group: SMF of 55–280 μT for 13 min on 12–15 consecutive days control group: ambient SMF of 38.6 ± 2.4 μT for 13 min on 12–15 consecutive days rats were exposed during daily learning session in the maze SMF generated by loudspeaker magnets installed beneath the maze rats were exposed and examined individually in the maze | food-operant behavior in a complicated problem environment (maze) and behavior in an open field test; drinking preference for ethanol/water | prolonged locomotor depression in the maze (no statistical analysis provided) such that exposed rats were unable to form food-operant behavior (impaired learning abilities); rats showed significantly less fear (p < 0.05) in open field; ethanol consumption was increased upon exposure (no statistical analysis provided) | same experimental setup as in Nikolskaya 1996 [42]; rats were treated differently: some rats in the exposure group received a "push" by the experimenter when motor activity inhibition occurred; also experiments with other magnetic flux densities (not relevant for this review) **study background** cognitive abilities, alcoholism |
| **Cardiovascular parameters** | | | | | |
| Ohkubo 1997 [44] (2nd tier) | rabbits (male) with an "ear chamber" attached to the ear lobe exposure group: n = 22, each animal was used as its own control (pre-exposure time) cage control group: n = 11 or 16 | exposure group: SMF of 1 mT for 10 min control group: GMF of approx. 20–30 μT rabbits were exposed individually (restrained in a fixing device) | cutaneous microcirculation/vasomotion in ear lobe | biphasic effects: when the vascular tone was low, SMF exposure induced vasoconstriction (p < 0.005), while vasodilation was induced by SMF exposure when the vascular tone was high (p < 0.001) | additional exposure groups (5 mT and 10 mT), not relevant for this review, data for these experiments not provided) **study background** potential clinical application of SMF for, e.g., relief from neck and shoulder stiffness and muscle fatigue |

*(Continued)*

**Table 1.** (Continued)

| First author (Year) Relevant groups (number of animals) | Exposed animals (sex) | Exposure | Endpoints | Outcome | Remarks |
|---|---|---|---|---|---|
| Okano 2001 [38] (1st tier) | rabbits (male) n = 34 (in total) 9 to 11 rabbits per group, animals in each group served as their own controls: group A: sham exposure vs. SMF of 1 mT group B: decreased blood pressure (induced by nicardipine) vs. decreased blood pressure + SMF of 1 mT group C: increased blood pressure (induced by L-NAME) vs. increased blood pressure + SMF of 1 mT group D (3 rabbits): each condition 30 trials in each group, 5 to 6 trials per rabbit | exposure conditions: SMF of 1 mT for 30 min control conditions: GMF inside the laboratory of approx. 46–47 µT rabbits were exposed individually (restrained in a fixing device) | (pharmacologically altered) blood pressure and microcirculation in a central artery of the ear lobe | **group A:** no significant effect on blood pressure (p-value not provided) nor on microcirculation (p ≥ 0.05) **group B:** SMF significantly suppressed the nicardipine-induced blood pressure reduction (p < 0.05) and the nicardipine-induced vasodilation (p < 0.05) **group C:** SMF significantly suppressed the L-NAME-induced blood pressure increase (p < 0.05) and the L-NAME-induced vasoconstriction (p < 0.05) | **study background** acute effects of SMF on pharmacologically altered blood pressure |
| Xu 2001 [46] (2nd tier) | BALB/c mice (male) exposure group (300 µT): n = 11, exposure group (1 mT): n = 15, control group: n = 20 | exposure group: SMF of 300 µT or 1 mT for 10 min control group: SMF of 46–47 µT for 10 min coils were aligned in a north/south direction mice were exposed individually | blood velocity in muscles | no statistically significant differences following 300 µT exposure (p ≥ 0.05), but increased blood velocity in the group exposed to 1 mT (p < 0.05) | mice were anesthetized; also extremely low frequency magnetic field exposure and other magnetic flux densities examined (not relevant for this review) **study background** fundamental research on blood pressure |
| Okano 2005 [37] (1st tier) | spontaneously hypertensive rats (male) exposure group: n = 20, control group: n = 20 | exposure group: mean SMF value of 550 µT (range 0.3–1.0 mT) continuously for 12 weeks control group: ambient SMF of approx. 50 µT continuously for 12 weeks SMF generated by permanent magnets rats were exposed individually | blood pressure, heart rate, skin blood flow; blood level of vasoactive substances (e.g., hormones) | **after 3 weeks of exposure:** statistically significantly reduced levels of angiotensin II (p < 0.01) and aldosterone (p < 0.001) **after 6 weeks of exposure:** statistically significantly reduced level of aldosterone (p < 0.05) **after 12 weeks of exposure:** no statistically significant differences in hormone levels (p ≥ 0.05) | all animals received an intraperitoneal saline injection (reason unclear); additional exposure group (not relevant for this review); partial replication of a previous study [48] **study background** fundamental research on blood pressure |

**Brain and nervous system**

| First author (Year) | Exposed animals (sex) | Exposure | Endpoints | Outcome | Remarks |
|---|---|---|---|---|---|
| Chance 1995 [47] (2nd tier) | rats (male and female) **1-month exposure:** 4 groups (male, female, male control, female control): n = 8–9 each **4-months exposure:** 4 groups (male, female, male control, female control): n = 6 each | exposure groups: SMF of 100 µT for 1 or 4 months (except for 15 min daily for animal care) control groups: ambient SMF of approx. 32 µT* (housed in the same room in a distance of 5 m to the coils) for 1 or 4 months main axis of the two pairs of coils were aligned in a North/South direction rats were simultaneously exposed in the same Helmholtz-coil apparatus but were each placed in a separate cage | neurotransmitter levels in the brain, amino acid levels in plasma, body weight | **after 1-month exposure:** statistically significant increase in hypothalamic levels of serotonin in male rats (p < 0.05) and statistically significant increase in striatal concentrations of 3-methoxytyramine in male and female rats (p < 0.01); maybe also effects on amino acid (taurine, glycine, and lysine) levels (acc. to text but not acc. to table, see "remarks") **after 4-months exposure:** no statistically significant differences | results taken from the text, results partially inconsistent between the text and the tables *magnetic flux density for control groups calculated from the provided data for single GMF components **study background** not outlined |
| Dincic 2018 [43] (2nd tier) | Wistar rats (male) exposure group (field orientation upward): n = 6, exposure group (field orientation downward): n = 6, control group: n = 6 | exposure groups: SMF of 1 mT (orientation upward or downward) for 50 days control group: naturally occurring SMF of 48 µT animals were likely housed and exposed in groups | enzyme activities of ATPases and acetylcholinesterase (AChE) and oxidative stress response (catalase activity, concentrations of hydroperoxides and malondialdehyde (MDA) in brain synaptosomes | significant increase of ATPases and acetylcholinesterase enzyme activities and MDA in both exposure groups (p < 0.05); significant decrease of catalase enzyme activity in downward orientation exposure group (p < 0.01) | **study background** effect of chronic SMF exposure, mechanism of action of SMF of different orientation on the nervous system |
| Zhang 2017 [39] (1st tier) | mice (male) implanted with microelectrode array in the hippocampal region exposure group: n = 7, control group: n = 7 | exposure group: SMF of 1 mT for 2h/day for 7 days control group: local GMF of approx. 40–50 µT for the same duration mice were exposed individually (restrained in an exposure tube) | working memory abilities, brain activity and brain histology | no statistically significant differences (p ≥ 0.05) | additional exposure group (50 Hz, not relevant for this review) **study background** working memory abilities in hippocampal region (primarily of 50 Hz magnetic fields) |

**3.3.1 Melatonin biosynthesis.** Two studies conducted by the same working group examined the potential effects of exposure to a weak SMF (52 μT and 72 μT) on melatonin biosynthesis in quails [40, 45]. The authors measured the enzyme activity of hydroxyindole-O-methyltransferase, which catalyzes the final reaction in melatonin biosynthesis in the retina and in the pineal gland, but the results of both studies were not consistent. In the experiment relevant for this review, Cremer-Bartels et al. [45] found that the retinal sensitivity was not affected by small magnetic field variations (SMF of 52 μT). However, for the pineal gland, Cremer-Bartels et al. [40] reported in their later study a statistically significant decrease in the enzyme activity after 20 and 60 minutes of exposure (SMF of 72 μT). The authors therefore suggested that weak SMF can influence the melatonin biosynthesis and thus might act as a "zeitgeber" for the circadian rhythm.

Both studies had several methodological limitations (e.g., adequate blinding of the research personnel was not reported) which might explain their inconsistent results. In particular, in their earlier study [45], the exposures were poorly described, i.e., the magnetic flux density was calculated but not measured. This is fairly problematic because the magnetic flux densities of the exposure and the control group were almost identical, i.e., 52 μT and 48 μT, respectively.

**3.3.2 Behavior.** Two studies by the same research group were identified that examined effects of exposure to SMF on behavior and cognitive abilities in rats. The experiments were done in a maze and included an appetitive conditioning task [41, 42]. In rats exposed to SMF of 55–280 μT, Nikolskaya et al. [42] and Nikolskaya and Echenko [41] observed both locomotor and emotional depression in the maze such that the rats were unable to form food-operant behavior. However, when the rats were removed from the maze and were observed in the open field, they demonstrated control levels of locomotor activity and learning abilities. The authors therefore concluded that cognitive processes are very sensitive to alterations of the SMF and suggested that SMF exposure can have an adverse effect in instances when the cognitive load is high, i.e., solving a food-rewarded learning task in a complicated problem environment. Additionally, in their later study, Nikolskaya and Echenko [41] observed a higher alcohol consumption in the exposed rats, but it remained unclear whether the difference between the exposed and the control group was statistically significant. The authors suggested that the increased alcohol consumption of the rats could have been caused by a combination of the SMF exposure and the demanding task in the maze.

For both studies [41, 42], there is concern whether the rats' behavior was rated by a non-blinded experimenter. Generally, observation of the animal behavior is a highly insensitive instrument because it relies on subjective judgments. Also, different treatment of the animals in the exposed and the control groups during the experiments suggests a role for some other factors besides SMF to explain the behavioral responses.

**3.3.3 Cardiovascular parameters.** Four studies–all conducted in the same laboratory–were concerned with effects of exposure to SMF on cardiovascular functioning, including blood velocity [46], cutaneous microcirculation [38, 44], blood pressure [37, 38], and heart rate plus levels of vasoactive peptide hormones [37]. The experiments were done in mice, rabbits, and rats. The applied magnetic flux densities ranged between 300 μT and 1 mT and exposure durations varied between 10 min and 12 weeks. The observed effects were dependent on the examined tissue, exposure duration, and magnetic flux density.

Locally applying a SMF of 1 mT to the cutaneous tissue in the rabbit ear lobe, Ohkubo et al. [44] and Okano and Ohkubo [38] consistently reported that exposure to SMF of 1 mT had a biphasic effect on microcirculation: when the vascular tone was low, vasoconstriction was induced while vasodilation was induced when the vascular tone was high. From additional results on pharmacologically altered blood pressure, Okano and Ohkubo [38] concluded that SMF exposure can influence modulations of $Ca^{2+}$ dynamics and alterations in

nitric oxide synthase activity. Similarly, Xu et al. [46] observed that exposure to SMF of 1 mT and 10 mT led to a statistically significant increase in the peak blood velocity in muscle capillaries of anesthetized mice. However, no such effect was seen for exposures to a weaker SMF of 300 µT. The authors suggested that a magnetic flux density of 1 mT can be considered as a threshold level above which enhanced muscle microcirculation in anesthetized mice is triggered.

When exposing mice to SMF for longer durations of 3 to 12 weeks, Okano and colleagues [37] provided some interesting insight into the variable effects of exposure on biochemical and physiological responses. This study was motivated by a previous study in which the authors found suppressed blood pressure elevation when applying higher SMF intensities [48]. For weak SMF (mean intensity of 550 µT), Okano et al. [37] reported reduced hormone levels (angiotensin II and aldosterone) after 3 and 6 weeks but not after 12 weeks of exposure. However, the observed hormone suppression after 3 and 6 weeks was neither reflected in a modification of blood pressure levels nor did it affect the development of hypertension. In contrast, exposure to a moderate SMF with a mean intensity of 2.8 mT suppressed and retarded the early stage development of hypertension. The results indicate that while modifications in biochemical responses might be prompted by lower SMF intensities at 550 µT, the physiological response, i.e., suppression and delay of blood pressure elevation, is only triggered at higher SMF intensities.

Several methodological limitations were identified in the study by Ohkubo and Xu [44] and the study by Xu et al. [46]. In particular, Ohkubo and Xu [44] did not provide information regarding the control of the exposure level and the control of potential confounders, which lowers the certainty that the reported effects are due to SMF exposure.

**3.3.4 Brain and nervous system.** Studies in three different laboratories investigated effects of exposure to SMF on the brain and nervous system. One study focused on effects of exposure on working memory abilities and brain activity [39]; the remaining two studies examined the influence of SMF on the levels of neurotransmitters [47] and on enzyme activities and oxidative stress in brain synaptosomes [43]. The experiments were done in mice and rats. The applied magnetic flux densities varied between 100 µT and 1 mT and the animals were exposed between 7 days (2 h/day) and 4 months (15 min/day). Chance and colleagues reported that exposure to SMF (100 µT for 1 month) increased the levels of serotonin and 3-methoxytyramine (a dopamine metabolite) in the brain of rats [47]. Prolongation of exposure to 4 months showed that the modifications in neurotransmitter levels and in circulating amino acids were transient and disappeared with continued exposure. However, it remained unclear from the interpretation of the results what significance they might have for health. Dincic and colleagues reported increases in enzyme activities (ATPase and acetylcholinesterase) and oxidative stress markers (malondialdehyde) in the brain of rats upon long-term exposure to differently oriented SMF of 1 mT [43]. While the authors concluded from their results that exposure to SMF might be a promising tool in the treatment of neuronal diseases, a possible mechanism for the observed alterations could not be proposed. In contrast, the study by Zhang et al. [39] did not indicate an effect of exposure to weak SMF on brain functioning. The authors found no changes in electrophysiological recordings from the hippocampus in mice that were exposed to SMF of 1 mT while performing a task that included working memory abilities.

As with studies related to other endpoints, we identified several methodological flaws in the studies by Chance et al. [47] and Dincic et al. [43] which lowered the confidence in the reported results.

## 4 Discussion

### 4.1 Summary of evidence

The aim of this systematic review was to analyze and evaluate the current knowledge on biological and health-related effects of exposure to weak SMF ($\leq$ 1 mT) in humans and vertebrates. For this purpose, we evaluated the outcomes of eleven experimental animal studies and critically appraised the individual internal validity of these studies. No human studies fulfilled our eligibility criteria.

Eight of the eleven reviewed studies reported effects of exposure to weak SMF. They were expressed as altered melatonin biosynthesis in quails [40], reduced locomotor activity in rats [41, 42], altered vasomotion [38, 44] and blood pressure [38] in rabbits, transient changes in blood pressure-related biochemical parameters in rats [37], transient changes in the level of neurotransmitters in rats [47], and increases in the enzyme activities in the rat brain [43]. The various effects were observed both upon short-term (e.g., 10 min [44]) and long-term (e.g., 4 months [47]) exposure. The results of the studies with positive findings are based on approximately 200 animals (rats, rabbits, or quails). No effects were reported in studies that exposed mice.

Given that SMF can interact with biological systems and may act, e.g., on electron spin interactions or exert forces on moving electric charges, it is possible that the reported effects indicate an association between exposure to weak SMF and biological functioning. However, due to the limited number of the included studies and the large heterogeneity in the study parameters but also because of partially inconsistent results and a lack of scientific rigor in most studies, the quality of evidence remains inadequate for drawing a conclusion for or against biological and health-related effects of exposure to SMF $\leq$ 1 mT for most endpoints. Two 1st tier and one 2nd tier studies revealed effects of exposure on several cardiovascular parameters [37, 38, 44] which provides some evidence for effects of exposure on the cardiovascular system. However, none of the effects reported in the reviewed studies have been confirmed in replication studies by independent investigators.

It should be noted that it remained largely unclear from the interpretation of the results whether the reported effects in the evaluated studies are beneficial or detrimental for health. However, it is unlikely that the reported effects of exposure to weak SMF pose a serious risk for health.

### 4.2 Limitations

A number of limitations need to be addressed when interpreting the results of our systematic analysis.

Because our systematic review focused on weak SMF, the data is not appropriate to draw any conclusions on the effects of exposure to SMF with higher magnetic flux densities as emitted, e.g., by MRI. Consequently, for conducting a comprehensive risk assessment of health outcomes from exposure to moderate and high SMF, also studies applying SMF > 1mT need to be systematically evaluated.

Also, there is a possible risk of publication bias in this line of research. Studies indicating no causal relation between SMF exposure and biological functioning were probably less likely to be published, thus potentially biasing the available literature.

The conclusions of this review are based on the studies identified by using the outlined search strategy. Because we only considered peer-reviewed articles published in English or German we may have missed potentially relevant articles published in other languages. Additionally, gray literature was not considered. It is also possible that relevant search terms for the

identification of articles could not be found in the title, abstract, or MeSH terms of some articles such that the searches in the EMF-Portal and PubMed did not return all potentially relevant articles.

Our eligibility criteria ruled out the evaluation of *in vitro* studies that provide insight into general mechanisms through which SMF can interact with biological systems. Also not considered in this review were studies investigating co-exposures or studies in which SMF exposures were attenuated with regard to the control group. Our conclusions may thus not apply to these types of studies.

Finally, due to the lack of sufficiently similar data, we could not perform quantitative analyses (e.g., meta-analysis) in this systematic review.

## 5 Conclusion

The reviewed studies display a great degree of heterogeneity with regard to the endpoints, the animal species examined, and the study parameters such that the currently available evidence regarding the potential for adverse effects of weak SMF is not sufficient to draw a firm conclusion. The conclusions of this review are thus consistent with those of former assessments. The WHO [17] noted that research on SMF exposure has often not been conducted systematically and lacked appropriate methodology and detailed information on exposure parameters. Other international commissions and review authors came to similar conclusions and criticized a lack of sufficient data for performing a risk assessment for exposure to SMF [49–51]. The strength of our review in comparison to former assessments is that we evaluated more recent studies, a larger number of studies addressing the effects of weak SMF (n = 11) than those considered, e.g., by WHO [17] (n = 5), and formally assessed the risk of bias in these studies.

Regarding the potential for adverse effects of SMF on biological functioning, former assessments concluded that any such effects are likely to be expected in the millitesla range and above. For example, the WHO [17] considers it probable that melatonin production may be suppressed upon exposure to moderate or high SMF intensities, but pointed to the inconsistent results between laboratories and therefore highlighted the need for further research. Further, the WHO [17], the Scientific Committee on Emerging and Newly Identified Health Risks [50], and ICNIRP [22] concluded that SMF exposures to moderate and high magnetic flux densities may induce effects on behavior and cardiovascular functions. However, the effects of SMF > 1 mT have not been analyzed in our review and should be systematically evaluated in future reviews.

Our analysis revealed that also weak SMF in the microtesla range may interact with biological systems. Eight of eleven reviewed studies reported effects of weak SMF on melatonin biosynthesis, locomotor activity, blood pressure regulation, brain enzyme activities, or neurotransmitter levels. However, based on the small number of available studies and on the assessment that most of the included studies lacked scientific rigor and homogeneity regarding study parameters, further research–including replication studies–is needed to evaluate in more detail the potential for effects of weak SMF on biological systems.

Since there is a rapid development in technologies that generate SMF, in particular with regards to proposals for new HVDC transmission lines or systems operating with batteries, such as electric cars, it is appropriate to assess the biological effects of exposure to a broad range of SMF intensities in rigorous and systematic research. A particular focus should be on the evaluation of the effects of exposure to SMF in humans. It has to be noted, however, that weak SMF as they are emitted from HVDC power lines or other technical applications, are partially in the range of the GMF, which could make it challenging to separate potential effects that are caused by exposures to the man-made sources from exposures to the GMF.

In planning any new experimental studies, we encourage researchers to use a well-controlled and validated exposure setting. For experiments involving exposures to weak SMF, precise measurements of the magnetic flux density should be obtained for the exposure group and also for the control/sham exposure group to control the level of background fields [52]. Detailed guidance on proper dosimetry in EMF research has been provided by Makinistian 2018 [52], Misakian 1993 [53], and Valberg 1995 [54]. In order to facilitate the comparison of exposures among studies and the synthesis of the results, we further encourage researchers to consider the reporting standards defined, e.g., in the ARRIVE guidelines for experimental animal studies [55], in the publication checklist by Hooijmans 2010 [56], or in the CONSORT statement for human clinical studies [57].

## Supporting information

**S1 Checklist.**
(DOC)

**S1 Table. Schematic overview for placement of individual human and animal studies in study quality categories (1 st tier, 2nd tier, 3rd tier).**
(DOCX)

**S2 Table. Excluded articles after full text screening (eligibility).**
(DOCX)

**S1 Data. PRISMA 2009 Checklist.**
(DOC)

**S1 Link.**
(DOCX)

**S2 Link.**
(DOCX)

## Acknowledgments

We thank the two reviewers for their valuable comments and suggestions which helped us to substantially improve the manuscript.

## Author Contributions

**Conceptualization:** Sarah Driessen, Dominik Stunder, Anne-Kathrin Petri.

**Data curation:** Sarah Driessen, David Graefrath, Anne-Kathrin Petri.

**Formal analysis:** Lambert Bodewein, Dagmar Dechent, David Graefrath, Anne-Kathrin Petri.

**Funding acquisition:** Sarah Driessen, Dominik Stunder.

**Investigation:** Sarah Driessen, Lambert Bodewein, Dagmar Dechent, Anne-Kathrin Petri.

**Methodology:** Sarah Driessen, Anne-Kathrin Petri.

**Project administration:** Sarah Driessen.

**Supervision:** Sarah Driessen, Thomas Kraus.

**Writing – original draft:** Sarah Driessen, Anne-Kathrin Petri.

**Writing – review & editing:** Sarah Driessen, Lambert Bodewein, Dagmar Dechent, Kristina
   Schmiedchen, Dominik Stunder.

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
