## [Decision Letter · Decision Letter 0]

8 Aug 2019

PONE-D-19-18270

Biological and health-related effects of weak static magnetic fields (< 1 mT) in humans and vertebrates: a systematic review.

PLOS ONE

Dear Mrs. Petri,

Thank you for submitting your manuscript to PLOS ONE. After careful consideration, we feel that it has merit but does not fully meet PLOS ONE’s publication criteria as it currently stands. Therefore, we invite you to submit a revised version of the manuscript that addresses the points raised during the review process.

ACADEMIC EDITOR: 

In this manuscript the authors performed a systematic review of the literature on biological and health-related effects of weak SMF (< 1 mT). The standardized method to perform a systematic review has been correctly applied.

Nevertheless, the manuscript presents several criticisms, as argued in details by both reviewer.

The main points to be taken into account are related to the choice to limit the analysis to SMF below 1 mT and to use only EMF Portal as source of literature search.

I hope you will find constructive the comments in performing a substantial revision before resubmit the revised version of the manuscript. Please, carefully address all the issues raised by the reviewers and provide the requested answers.

We would appreciate receiving your revised manuscript by Sep 22 2019 11:59PM. To enhance the reproducibility of your results, we recommend that if applicable you deposit your laboratory protocols in protocols.io, where a protocol can be assigned its own identifier (DOI) such that it can be cited independently in the future. For instructions see: http://journals.plos.org/plosone/s/submission-guidelines#loc-laboratory-protocols

We look forward to receiving your revised manuscript.

Kind regards,

Maria Rosaria Scarfi

Academic Editor

PLOS ONE

Additional Editor Comments (if provided):

In this manuscript the authors performed a systematic review of the literature on biological and health-related effects of weak SMF (< 1 mT).

The standardized method to perform a systematic review has been correctly applied.

Nevertheless, the manuscript presents several criticisms, as argued in details by both reviewer.

The main points to be taken into account are related to the choice to limit the analysis to SMF below 1 mT and to use only EMF Portal as source of literature search.

I hope you will find constructive the comments in performing a substantial revision before resubmit the revised version of the manuscript. Please, carefully address all the issues raised by the reviewers and provide the requested answers.

Reviewers' comments:

Reviewer's Responses to Questions

**Comments to the Author**

1. Is the manuscript technically sound, and do the data support the conclusions?

Reviewer #1: Partly

Reviewer #2: Partly

2. Has the statistical analysis been performed appropriately and rigorously? 

Reviewer #1: N/A

Reviewer #2: N/A

3. Have the authors made all data underlying the findings in their manuscript fully available?

Reviewer #1: Yes

Reviewer #2: Yes

4. Is the manuscript presented in an intelligible fashion and written in standard English?

Reviewer #1: Yes

Reviewer #2: Yes

5. Review Comments to the Author

Reviewer #1: The submitted manuscript is a systematic review of experimental human and animal studies addressing the health effects of exposure to weak (< 1 mT) static magnetic fields (SMF).

The article complies with PLOS One publication criteria. In particular, it adheres to the PRISMA reporting guidelines for systematic reviews. The data support the conclusions, and the suggested research needs are sensible.

However, some aspects of the review are unclear or debatable.

(1) The decision to focus on levels of SMF below one mT is not very clear. The geomagnetic field is in the order of tenths of mT, and the current international exposure limit for the public is 400 mT. Apparently, the most important argument was the increased risk of childhood leukemia in relation to exposure to weak extremely low frequency magnetic fields (ELF-MF) consistently observed in epidemiologic studies (Introduction, p. 4, lines 88-90). Is there any substantiated common mechanisms of biological interaction for static and ELF magnetic fields? Swanson and Kheifets (J Radiol Prot 2012; 32: 413-8) assessed whether the geomagnetic field appears to be an effect modifier in studies of alternating magnetic fields, finding some, but rather limited and not statistically significant, evidence for this. The review rationale should be revised and better justified.

(2) The Introduction section mentions various sources of everyday exposure to SMF. Ordering these sources by increasing level of exposure, and reporting these levels, would be helpful to the readers.

(3) The use of a single source for the literature search is a minor limitation, as EMF-Portal is a specialized database of scientific literature on electromagnetic fields. However, a PubMed search would have been a useful complement, allowing also to illustrate the coverage of EMF-Portal on the specific subject.

(4) In presenting the eligibility criteria, the Authors claim that “For practicability [For convenience?], we only considered studies in which the experimental group was exposed to a higher SMF exposure level than the control group/during sham exposure (GMF and background field) such that the GMF was sufficiently controlled as a possible confounder. Therefore, a magnetic flux density also had to be provided for the control group/sham condition.” (Page 5, lines 109-112). This criterion is a relevant one, and deals with the adequacy of the experimental exposure setup. Thus, it merits an explicit mention as a reason for exclusion in the study flow chart, with the related number of exclusion (see point 7 below).

(5) Many exclusion criteria are established (page 6, lines 117-125). Some of them are “standard” (e.g. reviews, editorials, commentaries, letters). Others are very specifc (e.g. studies with co-exposures; studies examining the influence of a geomagnetic storm/geomagnetic disturbances; studies simulating a space environment (field deprivation); studies examining the effect of an attenuated or altered GMF; studies dealing with magnetoreception). The Authors should provide a (concise) rationale behind these exclusion criteria.

(6) The Authors used the NTP-OAHT Risk-of Bias tool to assess of the internal validity of the examined studies (page 8, lines 167-191). Nevertheless, the choice not to consider as key-criteria the randomness of allocation (selection bias), and blinding of research personnel during the study (performance bias) is surprising in a systematic review of experimental studies. The meaning of tiers should be explained.

(7) The flow diagram of the literature search (Fig. 1) is not exactly in line with PRISMA guidelines; it is not sufficient to report the overall number of excluded studies; the numbers of studies excluded by specific reason should also be reported.

(8) A full list of the studies excluded (at the full-text analysis step), with reason and full reference, should be added (eventually as online supplementary material).

(9) Only 8 eligible studies were identified, published between 1983 and 2005. In principle, all these studies should have been included in the WHO hazard assessment of static fields (Environmental Health Criteria 232; 2006). Did the current review identify a larger/smaller number of “eligible” studies compared to the WHO monograph? The result of this cross-check should be reported.

(10) There is a mistake in Table 1, concerning the study by Okano et al. 2005 [31]; the Author claims that it was “unclear whether animals were housed and exposed individually or in groups”. This is wrong; the original paper specifies this aspect of the experimental conditions (“All animals were housed in the same room, with a 12 h light/dark cycle (lights on 07:00–19:00 h) at a temperature of 23.0+/-0.58 C, and a relative humidity of 50 +/-5%. Each animal was housed individually in a cage with free access to laboratory chow and tap water ad libitum").

(11) Replace “methodical” (7 occurrences) with “methodological”.

Reviewer #2: In this paper, the authors conducted a systematic review of the scientific literature regarding the assessment of biological effects in vivo (human and vertebrates) by exposure to weak (<1 mT) static magnetic fields (SMF). The authors adopted standardized methodologies for the literature search and reporting of results (PRISMA method, PECO strategy, analysis of risk of bias…). Even though I appreciate the effort in performing a systematic review of the literature, the paper presents several limitations that make it not recommendable for publication in its current form, as detailed in the following.

1. The main problem with this article is the lack of a clear rationale for limiting the analysis to studies dealing with SMF at magnetic induction levels below 1 mT. There are some intrinsic contradictions in the paper. As an example, in the introduction section the authors state that “… the aim of this systematic review was therefore to evaluate whether there is evidence that weak SMF (< 1 mT), as they occur e.g. near HVDC lines or batteries, can affect biological functioning in humans and vertebrates and cause adverse health effects”. Then, in the Discussion section, Summary of evidence, it is stated that: “Note that the experimental studies of animals exposed to SMF may have limited relevance to the magnetic field environment near to HVDC lines because SMF flux densities applied in most of the experimental studies reviewed were higher than the magnetic flux density under HVDC transmission lines.”. Therefore, first they refer to SMF from HVDC lines as one of the possible source of weak SMF to be considered for risk assessment, but then state that their conclusions do not apply to HVDC lines since the SMF in that case can be much higher. So, why not extending the search to higher SMF induction level, which may more likely occur in the real life? This would have increased the number of included studies and made their analysis more robust. As it is, the papers looks like an elegant exercise on how to conduct a systematic research, but useless if one wants to gain consistent information for risk assessment.

2. Again, in the introduction, the authors refer to papers dealing with correlation between childhood leukemia and exposure to ELF magnetic field. Specifically, they state that:

“In particular, an increase in the incidence rate of childhood leukemia has been consistently reported in epidemiological studies on exposure to ELF MF below the recommended limit values [19-23]. It has to be noted, however, that so far, no mechanism of action could be identified and that the reported increased incidence rate has not been confirmed by animal studies. Yet, the indicated association between weak ELF MF exposure and an increased risk for childhood leukemia led us to the question whether exposure to weak SMF may pose a risk to human health. Therefore, the aim of this systematic review was therefore to evaluate whether there is evidence that weak SMF (< 1 mT)…” It is difficult to get the correlation between these two concepts. The rationale for limiting the analysis to SMF below 1 mT is still lacking

3. Another questionable technical aspect is the search strategy. Did the authors perform a parallel search on another search engine than EMF Portal? I think that, by using more than one search term, they could have increased the sensitivity of their search strategy.

4. The whole Discussion and Conclusion sections present several repetitions of the same concept, i.e. the absence of sufficient data to draw firm conclusions for biological and health-related effects of exposure to weak SMF. I think this is a logical consequence of the choice to limit the analysis to SMF below 1 mT. I strongly encourage the authors to perform their analysis by enlarging at least SMF exposure conditions (above 1 mT) in the search strategy.

6. PLOS authors have the option to publish the peer review history of their article (what does this mean?). If published, this will include your full peer review and any attached files.

Reviewer #1: No

Reviewer #2: No

---

## [Author Response · Author response to Decision Letter 0]

15 May 2020

We are thankful to the two reviewers for their interest and time they spent on carefully reading our manuscript. Their feedback was valuable and very constructive. All comments and suggestions made by the reviewers were very helpful to improve the manuscript. The comments concerning the search strategy, the eligibility criteria of the articles to be included in this systematic review and the rationale for this evaluation have resulted in the most extensive revisions. 

Response to reviewer #1:

We are very thankful to the reviewer for his/her interest and the time he/she spent on carefully reading our manuscript. The comments helped us to substantially improve the manuscript. All points raised by the reviewer are addressed in a point-by-point fashion below. 

The submitted manuscript is a systematic review of experimental human and animal studies addressing the health effects of exposure to weak (< 1 mT) static magnetic fields (SMF).

The article complies with PLOS One publication criteria. In particular, it adheres to the PRISMA reporting guidelines for systematic reviews. The data support the conclusions, and the suggested research needs are sensible.

Response: Thank you very much for this positive statement. We have addressed all your comments below.

However, some aspects of the review are unclear or debatable.

(1) The decision to focus on levels of SMF below one mT is not very clear. The geomagnetic field is in the order of tenths of mT, and the current international exposure limit for the public is 400 mT. Apparently, the most important argument was the increased risk of childhood leukemia in relation to exposure to weak extremely low frequency magnetic fields (ELF-MF) consistently observed in epidemiologic studies (Introduction, p. 4, lines 88-90). Is there any substantiated common mechanisms of biological interaction for static and ELF magnetic fields? Swanson and Kheifets (J Radiol Prot 2012; 32: 413-8) assessed whether the geomagnetic field appears to be an effect modifier in studies of alternating magnetic fields, finding some, but rather limited and not statistically significant, evidence for this. The review rationale should be revised and better justified.

Response: The reviewer has raised an important aspect and we acknowledge that the rationale of our systematic review and our focus on publications with exposures to man-made SMF of ≤ 1 mT was not sufficiently outlined in the previous version of the manuscript. The primary reason to limit our inclusion criteria to articles applying SMF exposures ≤ 1 mT is that sources producing weak SMF are the most relevant for the public. To address the reviewer’s concerns as well as the concerns of reviewer #2, we restructured the introduction and now explain our rationale in more detail, particularly in the opening and the final paragraph of the introduction. To avoid misunderstanding, we removed the section on childhood leukemia associated with ELF magnetic fields. Instead, we added a more general section regarding the public discussion on possible health-related effects of exposure to electric, magnetic and electromagnetic fields following the section where we address the proposed limit values for SMF exposure.

Page 3, lines 47-62 

More recently, the installation of new high-voltage power lines (HVDC) [1, 2] and the introduction of novel technologies that produce electromagnetic fields (EMF), such as 5G networks [3], smart meters [4] and electric vehicles [5] have led to controversial discussions among the public, politicians, non-governmental organizations, and the industry about the benefits of these technologies and the possible risks of exposure to non-ionizing fields. Besides environmental impacts and legal requirements, the discussions have often centered on potential hazards to health from EMF. Public opposition often delays the roll-out of new technologies threatening the economy on various scales due to, e.g., delays in technical progress, potential international competitive disadvantages and serious financial losses. In Germany, both the launch of 5G networks and the plans for the build-out of the power grid, including four cross-country HVDC power lines have attracted growing public interest. New power lines are planned to transfer power generated by remote renewable sources (particularly wind power) to areas where the demand for energy is high. HVDC power lines can transport electricity over long distances with lower line losses compared to conventional alternating current power lines [6]. Although the intensity of static magnetic fields (SMF) emitted from HVDC lines is comparatively weak, there is public concern about possible health-related and environmental impacts of SMF produced in the vicinity of HVDC power lines.

(…)

Page 4, Lines 91-113

Exposure to SMF below the international limit values recommended by the International Commission on Non-Ionizing Radiation Protection (ICNIRP) [22], i.e., 400 mT for the public, are considered safe and they are not expected to pose a risk to health. However, the proposed exposure limits to electric, magnetic and electromagnetic fields are based on short-term (acute) health effects. This has led to strong public and scientific debate about the sparsely investigated effects of long-term exposure to low intensity EMF for the population. EMF sources that have attracted most attention were mobile phones and alternating current power lines (50/60 Hz) because it was discussed that long-term exposure to low intensity radiofrequency or extremely low frequency magnetic fields may increase the risk of cancer [28, 29]. 

The aim of this systematic review was therefore to evaluate whether there is evidence that weak SMF (≤ 1 mT), as they are produced, e.g., near HVDC lines and other man-made SMF sources of our daily life, can affect biological functioning in humans and vertebrates and cause adverse health effects. Formerly published reviews and reports on possible health risks from exposure to SMF were not conducted systematically and little information has been provided regarding the effects of exposure to weak SMF (≤ 1 mT). However, SMF sources that produce magnetic flux densities below 1 mT are the most relevant sources for public exposure. We collected and analyzed experimental in vivo studies on short-term and long-term biological and health-related effects of exposure to weak SMF. Our review is intended to critically appraise the internal validity of the published evidence, identify open research questions and support risk communication activities on the potential hazards of EMF. Although the likelihood of adverse health effects from exposure to weak SMF below the limit values was judged to be low, with the growing exposure to SMF produced by technical applications there is public and scientific demand for periodic evaluations of the current state of research in order to reassess and confirm the safety of weak SMF as they occur in daily life. 

(2) The Introduction section mentions various sources of everyday exposure to SMF. Ordering these sources by increasing level of exposure, and reporting these levels, would be helpful to the readers.

Response: The reviewer is right that ordering and reporting the sources by increasing level of exposure to SMF would be helpful to the reader. We have revised the paragraph as follows:

Page 4, lines 71-79

HVDC lines produce weak SMF between 22 µT and 38 µT in close proximity [6, 11, 12]. For hair dryers, stereo headsets, home sewing machines and electric clocks, it has been documented that their geometric mean SMFs are between 50 µT and 93 µT, depending on measurement distance and location [13]. Weak SMFs were measured inside hybrid technology cars (up to 0.95 mT) [14] and inside the driver’s cabin in direct current (DC) trains (about 1 mT) [15]. Moderate SMF may be produced by magnetic levitation train systems (up to 10 mT) [16] and in certain locations of aluminum production plants (60 mT) [17]. Under common exposure scenarios, magnet resonance imaging (MRI) workers are exposed to SMFs in the order of several hundreds of mT [18, 19] or even 2,890 mT (7 T scanner) in a research environment [20].

(3) The use of a single source for the literature search is a minor limitation, as EMF-Portal is a specialized database of scientific literature on electromagnetic fields. However, a PubMed search would have been a useful complement, allowing also to illustrate the coverage of EMF-Portal on the specific subject.

Response: We are thankful to the reviewer for this remark. Although PubMed is – besides other electronic databases – one of the major sources to search for relevant articles for inclusion in the literature inventory of the EMF-Portal, we performed an additional PubMed search to complement our literature search. Further, we extended the literature search in the EMF-Portal by adding several key words. We revised section 2.2. (“Information sources and literature search strategy”) and Figure 1 accordingly. Additionally, we provide all search strings in the supplementary data (S1 File, search string).

Page 7, lines 162-164

Relevant articles published through October 2019 were identified through electronic searches in PubMed (U.S. National Library of Medicine, National Institutes of Health) and in our highly specialized literature database EMF-Portal (www.emf-portal.org).

Page 8, lines 180-183

The search terms were related to exposures and included the following key words: static magnetic field(s), DC magnetic field(s), constant magnetic field(s), stationary magnetic field(s), steady magnetic field(s), magnetostatic field(s), high-voltage direct current, HVDC. The search strings and links to the electronic databases are provided in the S1 File (search string).

Compared to the initial search, we identified more than twice as many records through the updated and extended searches in PubMed and in the EMF-Portal. However, most of the identified articles had to be excluded after screening of title and abstract (see Figure 1). Nearly one quarter (n=1,008) of all articles were excluded because they were not relevant to our topic (exclusion reason: “not EMF/health-related”). Finally, the number of articles which fulfilled our inclusion criteria was similar to the number of articles we had included in the previous version of our manuscript. This confirms the coverage and the high degree of specialization of the EMF-Portal together with high search accuracy for relevant articles.

(4) In presenting the eligibility criteria, the Authors claim that “For practicability [For convenience?], we only considered studies in which the experimental group was exposed to a higher SMF exposure level than the control group/during sham exposure (GMF and background field) such that the GMF was sufficiently controlled as a possible confounder. Therefore, a magnetic flux density also had to be provided for the control group/sham condition.” (Page 5, lines 109-112). This criterion is a relevant one, and deals with the adequacy of the experimental exposure setup. Thus, it merits an explicit mention as a reason for exclusion in the study flow chart, with the related number of exclusion (see point 7 below).

Response: We agree with the reviewer that this exclusion criterion and the number of concerned articles should be provided more concisely in the flow diagram (Fig 1). A total of 15 studies were excluded because the “magnetic flux density of the control group/condition was not provided”. We added this information in more detail to the flow chart and further list all reasons and numbers of the excluded articles in the manuscript text (see also response to comment #7).

Page 10-11, lines 238-248

Of these, 323 articles were excluded for several reasons: magnetic flux density > 1 mT (n = 162), studies on (mechanisms of) magnetoreception (n = 58), other type of experimental study (n = 22, e.g., in vitro studies, studies on invertebrates or plants, dosimetric approach), magnetic flux density not provided (n = 17), magnetic flux density of the control group/condition not provided (n = 15), exposure conditions unclear (n = 8) or investigation of field deprivation/hypomagnetic field (n = 8). Other articles were excluded because they investigated co-exposures (n = 7), MRI (n = 6), a geomagnetic storm condition (n = 5) or because they had no SMF exposure condition (n = 5). Reviews, editorials, comments (n = 6), an article not written in English or German (n = 1), a non EMF health-related article (n = 1), and a non peer-reviewed study (n = 1) were also excluded. One further article had to be excluded because it lacked the description of the results for the exposure groups.

(5) Many exclusion criteria are established (page 6, lines 117-125). Some of them are “standard” (e.g. reviews, editorials, commentaries, letters). Others are very specifc (e.g. studies with co-exposures; studies examining the influence of a geomagnetic storm/geomagnetic disturbances; studies simulating a space environment (field deprivation); studies examining the effect of an attenuated or altered GMF; studies dealing with magnetoreception). The Authors should provide a (concise) rationale behind these exclusion criteria.

Response: Thank you for this recommendation. We agree that providing a rationale behind our exclusion criteria would be helpful to the reader. We revised the paragraph in section 2.1. (“Eligibility criteria”) as follows:

Pages 6-7, lines 136-160

Excluded were review articles, editorials, commentaries, unpublished or non peer-reviewed articles as well as studies on simulations and dosimetric or theoretical aspects. Also excluded were studies with co-exposures (e.g., as in MRI, which is a combination of exposures to SMF, radiofrequency electromagnetic fields and gradient magnetic fields or as in aluminum reduction plants, which are subject to multiple exposures such as heat and chemicals), because in these studies it may not be possible to separate a potential effect of exposure to SMF from the effects of the other exposure types.

Studies examining the influence of a geomagnetic storm or geomagnetic disturbances on health-related effects were excluded because they mainly investigate fluctuations of the GMF in the nanotesla range or experimentally simulate these fluctuations. The results of these studies preclude dosimetric considerations between magnetic flux densities and potential effects of exposure because the effect may be caused by the fluctuation itself rather than by a specific magnetic flux density. For this reason, these studies are not relevant for our systematic review.

Similarly, studies investigating an attenuated or hypomagnetic field such as, e.g., in a space environment, were not within the scope of our review. Because living organisms, including humans, are continuously exposed to the natural GMF, our research question was focused on man-made SMF that are superimposed on natural GMFs and not on the attenuation of GMFs. It has to be noted, however, that under specific circumstances the natural GMF may be subject to attenuation by man-made SMF.

As many species are able to perceive and orient to the GMF, magnetoreception and magnetic sensitivity were examined in a large number of studies. A great many of these studies investigated the effect of a variation in the magnetic flux density on magnetoreception or varied the inclination angle, the polarity or other environmental cues, such as light parameters. Because of their particular focus, studies on magnetoreception merit a separate evaluation and were therefore excluded from our review.

(6) The Authors used the NTP-OAHT Risk-of Bias tool to assess of the internal validity of the examined studies (page 8, lines 167-191). Nevertheless, the choice not to consider as key-criteria the randomness of allocation (selection bias), and blinding of research personnel during the study (performance bias) is surprising in a systematic review of experimental studies. The meaning of tiers should be explained.

Response: The reviewer has raised an important aspect. OHAT’s tiering approach is conceptually consistent with an approach outlined in the Cochrane handbook (https://training.cochrane.org/handbook/current) for reaching conclusions about the overall risk of bias in individual studies. The tiering approach defines key risk-of-bias criteria on a project-specific basis which are given the highest weight when it comes to placing of studies into quality categories (1st tier, 2nd tier, 3rd tier). These categories may then guide review authors when making judgments on the quality of evidence (confidence ratings) from all reviewed studies. 3rd tier studies, i.e., those that are at high risk-of-bias for all key criteria, are problematic in several key features of study design which raises serious concern about their internal validity. With 2nd tier studies, there is moderate confidence in the reported results while we can be very confident in the results of 1st tier studies. 

For observational studies, the OHAT approach proposes as key features “exposure assessment”, “outcome assessment” and “confounding variables”. However, for making confidence ratings on other study types, OHAT does not suggest key features of study design. As described in section 2.5 of our systematic review (“Study appraisal”), we have defined the following three key risk of bias criteria for this evaluation: (1) “Were experimental conditions identical across study groups?”, (2) “Can we be confident in the exposure characterization?” (i.e., exposure assessment), and (3) “Can we be confident in the outcome assessment?”. Two of the defined key elements (2 and 3) in our systematic review were derived from the OHAT approach for observational studies. However, the key criterion “confounding variables” does not apply for experimental studies because this issue is addressed through questions regarding randomization, allocation concealment and other threats to internal validity. Instead, we defined “Identical experimental conditions across study groups” as the third key feature of study design, because exposure-related confounders, such as background magnetic fields, heat, or vibration, that we evaluated in this category, are one of the most critical aspects when interpreting the results of EMF research. 

We completely agree with the reviewer’s rationale that randomness of allocation (selection bias) and blind rating of responses by the research personnel during the study (performance bias) are important features of experimental study designs. It has to be noted, however, that “blinding” is assessed for all three stages of the experimental procedures in the OHAT risk of bias rating tool: (1) during allocation concealment, (2) during study conduct, i.e., exposure and handling of the animals (Blinding of research personal) and (3) during outcome assessment. Because outcome assessment is defined as one of our key criteria, blinded procedures are actually considered in the ratings that were given the highest weight in determining the overall risk of bias of individual studies.

We added the meaning of tiers in section 2.5 (“Study appraisal”).

Pages 9, lines 218-220

This approach outlines a 3-tier system to rate study quality (1st tier: high confidence in the reported results, 2nd tier: moderate confidence in the reported results, or 3rd tier: low confidence in the reported results).

(7) The flow diagram of the literature search (Fig. 1) is not exactly in line with PRISMA guidelines; it is not sufficient to report the overall number of excluded studies; the numbers of studies excluded by specific reason should also be reported.

Response: We agree that it is useful information to report the numbers of studies excluded by specific reasons for all stages of assessment. The reason why we did not do this in the previous version of our manuscript is because many articles met more than one exclusion criteria during the first stage of assessment (screening of title and abstract). For example, a study that exposed cells lines to static magnetic fields of > 1mT was excluded, because neither the exposed system (in vitro) nor the magnetic flux density (> 1mT) fulfilled our inclusion criteria. Nevertheless, we defined for each study the most striking reason for exclusion and added the numbers of excluded studies to the specific reasons in Figure 1. Further, we added a note to section 3.1. (“Results – Study selection”) to make the reader aware of this reporting:

Page 10, lines 235-237

Note, that most of these studies were excluded because they met more than one exclusion criteria. For reasons of clarity, in the flow diagram, we only document the most striking reason for their exclusion (Fig 1).

Additionally, we report all exclusion reasons of the second stage of assessment (eligibility) in the manuscript text, section 3.1. (“Results – Study selection”) (see also Response to comment #4).

(8) A full list of the studies excluded (at the full-text analysis step), with reason and full reference, should be added (eventually as online supplementary material).

Response: Thank you for this advice. Providing a list of the excluded studies is an important aspect and was also recommended in the Cochrane Handbook for Systematic Reviews of Interventions (https://training.cochrane.org/handbook/current) and in the AMSTAR tool for assessing the methodological quality of systematic reviews (https://bmcmedresmethodol.biomedcentral.com/articles/10.1186/1471-2288-7-10 and https://www.bmj.com/content/358/bmj.j4008). We now provide in the supplementary data (Table S2) a list of all articles that were excluded at the second stage of assessment (eligibility) with full bibliographic data and reasons for their exclusion. Please note, that also at the second stage of assessment, some articles were excluded because they met more than one exclusion criteria. Same as with articles that were sorted out after the first stage of assessment, for reasons of clarity, we only document the most striking reason in Figure 1.

Page 10, lines 248-250

A list of all excluded articles including the bibliographic data and the reasons for their exclusion is provided in the Supplementary data (S2 Table).

(9) Only 8 eligible studies were identified, published between 1983 and 2005. In principle, all these studies should have been included in the WHO hazard assessment of static fields (Environmental Health Criteria 232; 2006). Did the current review identify a larger/smaller number of “eligible” studies compared to the WHO monograph? The result of this cross-check should be reported.

Response: We agree with the reviewer that it is an interesting aspect to compare the articles evaluated in our systematic review with the references included in the WHO Environmental Health Criteria 232 (2006). The WHO Environmental Health Criteria 232 (2006) includes approximately 600 publications on the effects of static electric and static magnetic fields of any field strength. Previously, we had already compared the inventory of this document with that of the EMF-Portal and published it in a research report (in German: Jahresberichte aus dem Forschungszentrum für Elektro-Magnetische Umweltverträglichkeit des Universitätsklinikums der RWTH Aachen, Bd. 15, 2013 (ISSN 1439-9261, https://www.ukaachen.de/fileadmin/files/global/user_upload/femu_forschungsbericht_2013_2.pdf; pages 43-44). All references of the WHO Environmental Health Criteria 232 that were missing in the EMF-Portal were added to the database, i.e., the EMF-Portal includes all references cited in the WHO document. They were thus retrieved by the systematic search for this review. The following table lists the articles that were eligible for our systematic review and compares the inclusion of articles between the WHO document and our review. Interestingly, our systematic search identified 6 additional articles that had not been considered in the WHO document. 

Table A: Comparison of studies included in WHO EHC 232 document and in our systematic review

 WHO EHC 232 (2006) Our systematic review (previous version) Our systematic review (current version)

Chance 1995 [47] not included included included

Cremer-Bartels 1983 [45] not included included included

Cremer-Bartels 1984 [40] not included included included

Dincic 2018 (1 mT) [43] not included not included included

Nikolskaya 1996 [42] not included included included

Nikolskaya 2002 [41] included included included

Ohkubo 1997 (1 mT) [44] included not included included

Okano 2001 (1 mT) [38] included not included included

Okano 2005 [37] included included included

Welker 1983 included included not included

Xu 2001 [46] included included included

Zhang 2017 (1 mT) [39] not included not included included

We have included the results of the WHO cross-check in section 5 (“Conclusion”).

Pages 22, lines 439-442

The strength of our review in comparison to former assessments is that we evaluated more recent studies, a larger number of studies addressing the effects of weak SMF (n = 11) than those considered, e.g., by WHO [17] (n = 5), and formally assessed the risk-of-bias in these studies.

The reviewer is right to point out that only few studies were included and evaluated in our review. A risk assessment may thus not be based on a solid ground. During the updated selection of potentially eligible studies, we noted that some studies applied SMF of exactly 1 mT that were previously excluded from our review because they fell out of the defined range of < 1 mT. Although field strengths of 1.0 mT are by definition classified as moderate field strengths, we decided to slightly modify our inclusion criterion for the magnetic flux density such that all studies applying SMF up to 1 mT were eligible for this review. Additionally, we updated our search and included any new studies published until the end of October 2019: 

Page 7, line 162

Relevant articles published through October 2019 were identified through (…)

The updated search and broadening of the inclusion criterion regarding the magnetic flux density returned four additional publications. We have changed all numbers and search periods where necessary. 

Also, we changed “< 1 mT” to “≤ 1 mT” wherever it was appropriate throughout the manuscript. 

Note that the study by Welker et al. (Welker HA, Semm P, Willig RP, Commentz JC, Wiltschko W, Vollrath L. Effects of an artificial magnetic field on serotonin N-acetyltransferase activity and melatonin content of the rat pineal gland. Exp Brain Res. 1983; 50(2-3):426-32. doi: 10.1007/BF00239209), which was evaluated in the previous version of our manuscript, was excluded from the current version because it did not meet any more our inclusion criteria (exclusion reason: (mechanisms of) magnetoreception). Welker et al. (1983) varied not only the magnetic flux density, but also the inclination angle in the exposure groups, i.e., it was not clear whether the observed effect was induced by the variation in the magnetic flux density or because of the variation in the inclination angle. Both in the previous and the revised version of the manuscript, we excluded other similar publications on magnetoreception during the study selection procedure.

The revised version of the systematic review now evaluates the results of 11 studies. With the updated search, we identified two more relevant studies on the effects of exposure to SMF on cardiovascular parameters (Ohkubo and Xu 1997 [44] and Okano and Ohkubo 2001 [38]) and two additional studies on the effects of exposure on the brain and nervous system (Dincic et al. 2018 [43] and Zhang et al. 2017 [39]) (see also Table A).

The study parameters and results of the newly included studies are detailed in Table 1. The results and the interpretations of the reported data are further described in sections 3.3.3 (“Cardiovascular parameters”) and 3.3.4 (“Brain and nervous system”). The risk-of-bias assessment for these studies is presented in the revised Figure 2. 

Additionally, we revised and restructured the results section. In particular, we now describe the results of individual studies more concisely and compare them when possible. Note that we removed redundant information that is already presented in Table 1 and Figure 2. 

(10) There is a mistake in Table 1, concerning the study by Okano et al. 2005 [31]; the Author claims that it was “unclear whether animals were housed and exposed individually or in groups”. This is wrong; the original paper specifies this aspect of the experimental conditions (“All animals were housed in the same room, with a 12 h light/dark cycle (lights on 07:00–19:00 h) at a temperature of 23.0+/-0.58 C, and a relative humidity of 50 +/-5%. Each animal was housed individually in a cage with free access to laboratory chow and tap water ad libitum").

Response: Thank you very much for bringing the mistake to our attention. This is right and we apologize for the incorrect description of the study by Okano et al. (2005). We revised it accordingly in Table 1.

(11) Replace “methodical” (7 occurrences) with “methodological”.

Response: Agreed. We have changed wording throughout the manuscript. 

Response to reviewer #2:

We are very thankful to the reviewer for his/her interest and the time he/she spent on carefully reading our manuscript. The comments helped us to substantially improve the manuscript. All points raised by the reviewer are addressed in a point-by-point fashion below.

In this paper, the authors conducted a systematic review of the scientific literature regarding the assessment of biological effects in vivo (human and vertebrates) by exposure to weak (<1 mT) static magnetic fields (SMF). The authors adopted standardized methodologies for the literature search and reporting of results (PRISMA method, PECO strategy, analysis of risk of bias…). Even though I appreciate the effort in performing a systematic review of the literature, the paper presents several limitations that make it not recommendable for publication in its current form, as detailed in the following.

1. The main problem with this article is the lack of a clear rationale for limiting the analysis to studies dealing with SMF at magnetic induction levels below 1 mT. There are some intrinsic contradictions in the paper. As an example, in the introduction section the authors state that “… the aim of this systematic review was therefore to evaluate whether there is evidence that weak SMF (< 1 mT), as they occur e.g. near HVDC lines or batteries, can affect biological functioning in humans and vertebrates and cause adverse health effects”. Then, in the Discussion section, Summary of evidence, it is stated that: “Note that the experimental studies of animals exposed to SMF may have limited relevance to the magnetic field environment near to HVDC lines because SMF flux densities applied in most of the experimental studies reviewed were higher than the magnetic flux density under HVDC transmission lines.”. Therefore, first they refer to SMF from HVDC lines as one of the possible source of weak SMF to be considered for risk assessment, but then state that their conclusions do not apply to HVDC lines since the SMF in that case can be much higher. So, why not extending the search to higher SMF induction level, which may more likely occur in the real life? This would have increased the number of included studies and made their analysis more robust. As it is, the papers looks like an elegant exercise on how to conduct a systematic research, but useless if one wants to gain consistent information for risk assessment.

Response: The reviewer has raised an important aspect and we acknowledge that the rationale of our systematic review and our focus on publications with exposures to man-made SMF of ≤ 1 mT was not sufficiently outlined in the previous version of the manuscript. The primary reason to limit our inclusion criteria to articles applying SMF exposures ≤ 1 mT is that sources producing weak SMF are the most relevant for the public. To address the reviewer’s concerns as well as the concerns of reviewer #1, we restructured the introduction and now explain our rationale in more detail in the opening and the final paragraph of the introduction. 

Page 3, lines 47-62 

More recently, the installation of new high-voltage power lines (HVDC) [1, 2] and the introduction of novel technologies that produce electromagnetic fields (EMF), such as 5G networks [3], smart meters [4] and electric vehicles [5] have led to controversial discussions among the public, politicians, non-governmental organizations, and the industry about the benefits of these technologies and the possible risks of exposure to non-ionizing fields. Besides environmental impacts and legal requirements, the discussions have often centered on potential hazards to health from EMF. Public opposition often delays the roll-out of new technologies threatening the economy on various scales due to, e.g., delays in technical progress, potential international competitive disadvantages and serious financial losses. In Germany, both the launch of 5G networks and the plans for the build-out of the power grid, including four cross-country HVDC power lines have attracted growing public interest. New power lines are planned to transfer power generated by remote renewable sources (particularly wind power) to areas where the demand for energy is high. HVDC power lines can transport electricity over long distances with lower line losses compared to conventional alternating current power lines [6]. Although the intensity of static magnetic fields (SMF) emitted from HVDC lines is comparatively weak, there is public concern about possible health-related and environmental impacts of SMF produced in the vicinity of HVDC power lines.

(…)

Page 4, Lines 91-113

Exposure to SMF below the international limit values recommended by the International Commission on Non-Ionizing Radiation Protection (ICNIRP) [22], i.e., 400 mT for the public, are considered safe and they are not expected to pose a risk to health. However, the proposed exposure limits to electric, magnetic and electromagnetic fields are based on short-term (acute) health effects. This has led to strong public and scientific debate about the sparsely investigated effects of long-term exposure to low intensity EMF for the population. EMF sources that have attracted most attention were mobile phones and alternating current power lines (50/60 Hz) because it was discussed that long-term exposure to low intensity radiofrequency or extremely low frequency magnetic fields may increase the risk of cancer [28, 29]. 

The aim of this systematic review was therefore to evaluate whether there is evidence that weak SMF (≤ 1 mT), as they are produced, e.g., near HVDC lines and other man-made SMF sources of our daily life, can affect biological functioning in humans and vertebrates and cause adverse health effects. Formerly published reviews and reports on possible health risks from exposure to SMF were not conducted systematically and little information has been provided regarding the effects of exposure to weak SMF (≤ 1 mT). However, SMF sources that produce magnetic flux densities below 1 mT are the most relevant sources for public exposure. We collected and analyzed experimental in vivo studies on short-term and long-term biological and health-related effects of exposure to weak SMF. Our review is intended to critically appraise the internal validity of the published evidence, identify open research questions and support risk communication activities on the potential hazards of EMF. Although the likelihood of adverse health effects from exposure to weak SMF below the limit values was judged to be low, with the growing exposure to SMF produced by technical applications there is public and scientific demand for periodic evaluations of the current state of research in order to reassess and confirm the safety of weak SMF as they occur in daily life. 

Additionally, to avoid misunderstanding, we removed the following sentence in section 4.1. (“Summary of evidence”):

Note that the experimental studies of animals exposed to SMF may have limited relevance to the magnetic field environment near to HVDC lines because SMF flux densities applied in most of the experimental studies reviewed were higher than the magnetic flux density under HVDC transmission lines.

2. Again, in the introduction, the authors refer to papers dealing with correlation between childhood leukemia and exposure to ELF magnetic field. Specifically, they state that:

“In particular, an increase in the incidence rate of childhood leukemia has been consistently reported in epidemiological studies on exposure to ELF MF below the recommended limit values [19-23]. It has to be noted, however, that so far, no mechanism of action could be identified and that the reported increased incidence rate has not been confirmed by animal studies. Yet, the indicated association between weak ELF MF exposure and an increased risk for childhood leukemia led us to the question whether exposure to weak SMF may pose a risk to human health. Therefore, the aim of this systematic review was therefore to evaluate whether there is evidence that weak SMF (< 1 mT)…” It is difficult to get the correlation between these two concepts. The rationale for limiting the analysis to SMF below 1 mT is still lacking

Response: The reviewer is right and we agree that we did not sufficiently outline the rationale for our systematic review. Please, see response to comment #1 how we addressed this concern. 

We removed the section on childhood leukemia associated with ELF magnetic fields. Instead, we added a more general section regarding the public discussion on possible health-related effects of exposure to electric, magnetic and electromagnetic fields following the section where we address the proposed limit values for SMF exposure (see above and pages 4-5, lines 93-99).

3. Another questionable technical aspect is the search strategy. Did the authors perform a parallel search on another search engine than EMF Portal? I think that, by using more than one search term, they could have increased the sensitivity of their search strategy.

Response: We are thankful to the reviewer for this remark. Although PubMed is – besides other electronic databases – one of the major sources to search for relevant articles for inclusion in the literature inventory of the EMF-Portal, we performed an additional PubMed search to complement our literature search. Further, we extended the literature search in the EMF-Portal by adding several key words. We revised section 2.2. (“Information sources and literature search strategy”) and Figure 1 accordingly. Additionally, we provide all search strings in the Supplementary data (S1 File, search string).

Page 7, lines 162-164

Relevant articles published through October 2019 were identified through electronic searches in PubMed (U.S. National Library of Medicine, National Institutes of Health) and in our highly specialized literature database EMF-Portal (www.emf-portal.org).

Page 8, lines 180-183

The search terms were related to exposures and included the following key words: static magnetic field(s), DC magnetic field(s), constant magnetic field(s), stationary magnetic field(s), steady magnetic field(s), magnetostatic field(s), high-voltage direct current, HVDC. The search strings and links to the electronic databases are provided in the S1 File (search string).

Compared to the initial search, we identified more than twice as many records through the updated and extended searches in PubMed and in the EMF-Portal. However, most of the identified articles had to be excluded after screening of title and abstract (see Figure 1). Nearly one quarter (n=1,008) of all articles were excluded because they were not relevant to our topic (exclusion reason: “not EMF/health-related”). Finally, the number of articles which fulfilled our inclusion criteria was similar to the number of articles we had included in the previous version of our manuscript. This confirms the coverage and the high degree of specialization of the EMF-Portal together with high search accuracy for relevant articles.

4. The whole Discussion and Conclusion sections present several repetitions of the same concept, i.e. the absence of sufficient data to draw firm conclusions for biological and health-related effects of exposure to weak SMF. I think this is a logical consequence of the choice to limit the analysis to SMF below 1 mT. I strongly encourage the authors to perform their analysis by enlarging at least SMF exposure conditions (above 1 mT) in the search strategy.

Response: The reviewer is right to point out that only few studies were included and evaluated in our review. A risk assessment may thus not be based on a solid ground. During the updated selection of potentially eligible studies, we noted that some studies applied SMF of exactly 1 mT that were previously excluded from our review because they fell out of the defined range of < 1 mT. Although field strengths of 1.0 mT are by definition classified as moderate field strengths, we decided to slightly modify our inclusion criterion for the magnetic flux density such that all studies applying SMF up to 1 mT were eligible for this review. Additionally, we updated our search and included any newly published study until the end of October 2019.

Page 7, line 162

Relevant articles published through October 2019 were identified through (…)

The updated search and broadening of the inclusion criterion regarding the magnetic flux density returned four additional publications. We have changed all numbers and search periods where necessary. 

Also, we changed “< 1 mT” to “≤ 1 mT” wherever it was appropriate throughout the manuscript. 

Note that the study by Welker et al. (Welker HA, Semm P, Willig RP, Commentz JC, Wiltschko W, Vollrath L. Effects of an artificial magnetic field on serotonin N-acetyltransferase activity and melatonin content of the rat pineal gland. Exp Brain Res. 1983; 50(2-3):426-32. doi: 10.1007/BF00239209), which was evaluated in the previous version of our manuscript, was excluded from the current version because it did not meet any more our inclusion criteria (exclusion reason: (mechanisms of) magnetoreception). Welker et al. (1983) varied not only the magnetic flux density, but also the inclination angle in the exposure groups, i.e., it was not clear whether the observed effect was induced by the variation in the magnetic flux density or because of the variation in the inclination angle. Both in the previous and the revised version of the manuscript, we excluded during the study selection process other similar publications on magnetoreception.

The revised version of the systematic review now evaluates the results of 11 studies. With the updated search, we identified two more relevant studies on the effects of exposure to SMF on cardiovascular parameters (Ohkubo and Xu 1997 [44] and Okano and Ohkubo 2001 [38]) and two additional studies on the effects of exposure on the nervous system (Dincic et al. 2018 [43] and Zhang et al. 2017 [39]).

The study parameters and results of the newly included studies are detailed in Table 1. The results and the interpretations of the reported data are further described in sections 3.3.3 (“Cardiovascular parameters”) and 3.3.4 (“Brain and nervous system”). The risk-of-bias assessment for these studies is presented in the revised Figure 2. 

Additionally, we revised and restructured the results section. In particular, we now describe the results of individual studies more concisely and compare them when possible. Note that we removed redundant information that is already presented in Table 1 and Figure 2.

---

## [Decision Letter · Decision Letter 1]

20 Feb 2020

Biological and health-related effects of weak static magnetic fields (≤ 1 mT) in humans and vertebrates: a systematic review.

PONE-D-19-18270R1

Dear Dr. Driessen,

We are pleased to inform you that your manuscript has been judged scientifically suitable for publication and will be formally accepted for publication once it complies with all outstanding technical requirements.

With kind regards,

Maria Rosaria Scarfi

Academic Editor

PLOS ONE

Additional Editor Comments (optional):

The manuscript has been significantly improved and the authors provided adequate answers to the reviewers. I recommend the publication of the manuscript

Reviewers' comments:

Reviewer's Responses to Questions

**Comments to the Author**

1. If the authors have adequately addressed your comments raised in a previous round of review and you feel that this manuscript is now acceptable for publication, you may indicate that here to bypass the “Comments to the Author” section, enter your conflict of interest statement in the “Confidential to Editor” section, and submit your "Accept" recommendation.

Reviewer #1: All comments have been addressed

Reviewer #2: All comments have been addressed

2. Is the manuscript technically sound, and do the data support the conclusions?

Reviewer #1: Yes

Reviewer #2: Yes

3. Has the statistical analysis been performed appropriately and rigorously? 

Reviewer #1: N/A

Reviewer #2: Yes

4. Have the authors made all data underlying the findings in their manuscript fully available?

Reviewer #1: Yes

Reviewer #2: Yes

5. Is the manuscript presented in an intelligible fashion and written in standard English?

Reviewer #1: Yes

Reviewer #2: Yes

6. Review Comments to the Author

Reviewer #1: I thank the Authors for appreciating my comments and suggestions of changes.

I am happy and honored to have collaborated with them on a constructive and effective peer review process.

Reviewer #2: THe authors have adquately addressed the issues raised in the first revision round. The rationale behind the choice of considering papers dealing with exposures to magnetic induction levels below 1 mT has been clarified. The paper is now recommendable for publication in this reviewer's opinion

7. PLOS authors have the option to publish the peer review history of their article (what does this mean?). If published, this will include your full peer review and any attached files.

Reviewer #1: Yes: Susanna Lagorio (MD, PhD, Senior Researcher), Istituto Superiore di Sanità (National Institute of Health) - Department of Oncology and Molecular Medicine, Rome, Italy

Reviewer #2: No

---

## [Editor Report · Acceptance letter]

15 May 2020

PONE-D-19-18270R1 

Biological and health-related effects of weak static magnetic fields (≤ 1 mT) in humans and vertebrates: a systematic review. 

Dear Dr. Driessen:

I am pleased to inform you that your manuscript has been deemed suitable for publication in PLOS ONE. Congratulations! Your manuscript is now with our production department. 

With kind regards,

on behalf of

Dr. Maria Rosaria Scarfi 

Academic Editor

PLOS ONE